# Synbiotic Supplementation Attenuates Doxorubicin-Induced Oxidative Stress and Inflammation in the Gut-Heart Axis of Chemotherapy-Treated Mice

**DOI:** 10.3390/ijms26115136

**Published:** 2025-05-27

**Authors:** Chi-Feng Cheng, Thi Kim Ngan Nguyen, Szu-Chuan Shen, Bo-Yu Chen, Yeh-B. Wu, Hui-Ju Liang, Chung-Hsin Wu

**Affiliations:** 1Department of Oncology, Ren Ai Branch, Taipei City Hospital, Taipei 10629, Taiwan; draculachi@yahoo.com.tw; 2School of Life Science, National Taiwan Normal University, Taipei 11677, Taiwan; marynguyen@ntnu.edu.tw (T.K.N.N.); scs@ntnu.edu.tw (S.-C.S.); boyu000706@gmail.com (B.-Y.C.); 3ARJIL Pharmaceuticals LLC, Hsinchu City 30013, Taiwan; ybw333@arjilbio.com (Y.-B.W.); kathy@arjilbio.com (H.-J.L.)

**Keywords:** synbiotics, doxorubicin, cancer chemotherapy, inflammation, mice

## Abstract

The gut microbiome supports immune health and influences gut and heart functions through the gut-heart axis. Synbiotics (SBT), combining probiotics and prebiotics, help restore microbiome balance. Chemotherapy often disrupts this balance, leading to adverse effects on the gut and heart. This study explores the potential of SBT supplementation in reducing heart and gut inflammation caused by doxorubicin (DOX) chemotherapy. The gut microbiome plays a vital role in immune health, and metabolites produced by gut bacteria contribute to physiological functions through the gut-heart axis. Chemotherapy drugs often disrupt these processes, leading to adverse effects on internal organs. Using 24 ICR male mice divided into four groups, the experiment assessed the impact of SBT on DOX-induced damage. Results indicated that DOX treatment significantly worsened survival rates, physical performance, heart function, and gut microbiome stability. However, co-treatment with SBT improved these markers, suggesting that SBT may help mitigate chemotherapy-induced side effects in cancer patients.

## 1. Introduction

Doxorubicin (DOX) has demonstrated significant efficacy in cancer treatment, yet its adverse side effects remain a major challenge in clinical application. The toxicity of DOX primarily affects the heart, gastrointestinal tract, bone marrow, and skin, significantly impacting patients’ quality of life and treatment adherence [1]. Cardiotoxicity is one of the most challenging adverse reactions of DOX treatment. DOX can cause myocardial cell damage, such as electrocardiogram abnormalities, and may even lead to heart failure. Studies have shown that DOX-induced cardiotoxicity is mainly due to oxidative stress reactions caused by the generation of reactive oxygen species (ROS), which damage mitochondrial function in myocardial cells and induce apoptosis [2]. Therefore, close monitoring of cardiac function in patients using DOX and timely cardiac protective measures, such as the use of antioxidants or cardioprotective drugs, are necessary in clinical application. Additionally, the impact of DOX on the gastrointestinal tract should not be overlooked. During chemotherapy, patients often experience symptoms such as vomiting, diarrhea, nausea, and oral ulcers. These side effects severely affect patients’ diet and nutritional intake, subsequently impacting the treatment’s efficacy and quality of life. DOX-induced gastrointestinal adverse reactions may have direct toxic effects or be related to disturbances in the gut microbiota [3]. Common measures to alleviate gastrointestinal adverse reactions include the use of antiemetics, anti-inflammatory drugs, and probiotics to improve gastrointestinal function and reduce discomfort symptoms. Besides pharmaceutical interventions, nutritional support and psychological counseling are important adjunctive treatments that help patients better cope with the adverse reactions brought by chemotherapy [4].

Synbiotics (SBT) are mixtures of live microorganisms and substrates selectively utilized by host microorganisms, which provide health benefits to the host. According to the latest definition from the International Scientific Association for Probiotics and Prebiotics (ISAPP), SBT are not merely a simple combination of probiotics and prebiotics but can be categorized into two types: (1) complementary SBT: these contain probiotics and prebiotics, where each component meets the minimum standards for probiotics or prebiotics and functions independently; and (2) synergistic SBT: the substrate is designed to be selectively utilized by the co-administered live microorganisms, thereby generating health benefits. A recent research study reported that SBT can enhance the immune system by regulating the balance of gut microbiota and reduce the gastrointestinal side effects induced by chemotherapy [5]. Thus, SBT supplementation may help to mitigate doxorubicin-induced oxidative stress and inflammation. During DOX chemotherapy, the adjunctive role of SBT has gradually gained attention. Studies have shown that SBT can alleviate gastrointestinal discomfort caused by chemotherapy, such as vomiting, nausea, and diarrhea, and enhance the repair of intestinal barrier function [6]. SBT can regulate gut microbiota, inhibit the proliferation of harmful bacteria, and maintain the balance of the gut microecosystem. Gut health significantly influences overall immune function. By enhancing the gut immune barrier function, SBT reduce the inflammatory response in the gut induced by chemotherapy, thereby alleviating patients’ gastrointestinal discomfort symptoms. This helps improve patients’ treatment adherence and quality of life [7]. SBT also promote the repair and regeneration of the intestinal mucosa. Chemotherapy drugs often damage the intestinal mucosa, leading to impaired intestinal barrier function, which can cause diarrhea and infections. SBT accelerate mucosal barrier recovery by promoting the proliferation and repair of intestinal mucosal cells, effectively reducing chemotherapy-induced intestinal injury [8]. They can also modulate the host immune response and enhance the body’s anti-infection capacity, further protecting gut health. Additionally, SBT show potential in alleviating systemic side effects induced by chemotherapy. Research has found that SBT can reduce systemic inflammatory responses caused by chemotherapy by regulating the immune system, thereby alleviating symptoms such as fatigue, loss of appetite, and immunosuppression [9]. These effects not only help improve patients’ quality of life but also may enhance the anticancer efficacy of chemotherapy by improving overall health status. As an adjunctive therapeutic approach, SBT show broad application prospects in reducing gastrointestinal and systemic adverse reactions induced by DOX chemotherapy. By regulating gut microbiota, promoting intestinal mucosal repair, and enhancing immune function, SBT are expected to improve patients’ treatment adherence and quality of life while enhancing the anticancer efficacy of DOX. Future research should further explore the specific applications of SBT in different types and stages of cancer treatment to provide more comprehensive and personalized treatment plans for cancer patients [10].

This study aims to comprehensively explore the efficacy and adverse side effects of DOX in cancer chemotherapy and to further investigate the potential of SBT as an adjuvant therapy. Through systematic experimental research and clinical trials, we provide cancer patients with more effective and safe treatment options, thereby improving patients’ treatment outcomes and lifestyle. Ultimately, this study hopes to promote the application of SBT in cancer treatment and provide new treatment options and scientific basis for clinical practice.

## 2. Results

### 2.1. Effects of SBT on Survival Rate and Body Weight in DOX-Treated Mice

Figure 1A evaluates the survival rate differences among groups using the Kaplan–Meier test. The results show that the sham and SBT groups maintained 100% survival throughout the experimental period, indicating no adverse effects from treatment. In contrast, the DOX group exhibited a sharp decline in survival starting from week 3, with survival dropping to approximately 20% by week 5. This suggests that DOX has a significant negative impact on survival. The SBT + DOX group showed a decrease in survival beginning in week 4, reaching around 60% by week 5. Compared to DOX alone, SBT appear to provide partial protective effects, mitigating the decline in survival.

Furthermore, Figure 1B illustrates a notable weight reduction in the DOX group, while the SBT + DOX group experienced a relatively smaller decrease. This trend is consistent with the survival rate findings, further supporting the protective potential of SBT against DOX toxicity.

### 2.2. Effects of SBT on Motor Dysfunction Induced by DOX Treatment in Mice

In this study, a rotarod test and treadmill were employed to assess motor coordination and physical endurance in ICR mice (Figure 2). Figure 2A highlights the latency to fall on the rotarod among mice subjected to sham, SBT, DOX, and SBT + DOX treatments over a three-week period. The results demonstrated that the latency to fall in the DOX-treated group was significantly shorter compared to the sham and SBT groups (*p* < 0.01). However, the latency to fall in the SBT + DOX-treated group was significantly prolonged compared to the DOX group (*p* < 0.01), indicating that SBT treatment helps mitigate DOX-induced motor coordination impairments.

Similarly, Figure 2B illustrates the running time on the treadmill among mice treated with sham, SBT, DOX, and SBT + DOX for three consecutive weeks. The DOX-treated mice exhibited a significantly shorter running time compared to the sham and SBT groups (*p* < 0.01). Conversely, the running time in the SBT + DOX group was significantly longer than that of the DOX group (*p* < 0.01). These findings suggest that SBT treatment effectively alleviates the reduction in exercise capacity and muscle endurance caused by DOX treatment in ICR mice.

### 2.3. Effects of SBT on Depressive Behavior and Weakened Grip Strength Induced by DOX Treatment in Mice

In this study, a forced swimming test was conducted to evaluate depressive behaviors in ICR mice following sham, SBT, DOX, and SBT + DOX treatments over three consecutive weeks (Figure 3A). The results showed that the immobility time during forced swimming in the DOX-treated group was significantly shorter than that of the sham and SBT groups (*p* < 0.01). However, the immobility time in the SBT + DOX-treated group was significantly longer than in the DOX group (*p* < 0.01), indicating that SBT treatment can effectively mitigate DOX-induced depressive behaviors in ICR mice. Additionally, grip strength was assessed in ICR mice subjected to the same treatments over the same period (Figure 3B). The results revealed that grip strength in the DOX-treated group was significantly weaker than that in the sham and SBT groups (*p* < 0.01). Conversely, the grip strength of the SBT + DOX-treated mice was significantly higher than that of the DOX group (*p* < 0.01), suggesting that SBT treatment can effectively alleviate grip strength impairments caused by DOX treatment in ICR mice.

### 2.4. Effects of SBT on Cold Sensitivity Induced by DOX Treatment in Mice

Patients undergoing cancer chemotherapy often experience heightened sensitivity to cold stimulation. Figure 4 illustrates the cold sensitivity scores of ICR mice following sham, SBT, DOX, and SBT + DOX treatments over three consecutive weeks. The results showed that the cold sensitivity scores of the DOX-treated mice were significantly higher than those of the sham and SBT groups (*p* < 0.01). In contrast, the scores of the SBT + DOX-treated mice were significantly lower than those of the DOX group (*p* < 0.01). These findings suggest that SBT treatment can effectively mitigate the cold sensitivity induced by DOX treatment in ICR mice.

### 2.5. Effects of SBT on Intestinal Dysfunction Induced by DOX Treatment in Mice

Fecal pellets were collected from ICR mice. The analysis focused on mice treated with sham, SBT, DOX, and SBT + DOX over three consecutive weeks. Figure 5 reveals that fecal pellet numbers in the DOX-treated group were notably lower than those in the sham and SBT groups. In contrast, the SBT + DOX group displayed higher fecal pellet numbers compared to the DOX group. These findings suggest that SBT treatment can counteract the reduction in fecal pellet count caused by DOX.

We quantitatively compared pathological changes in small intestine samples using the tissue lesion score (SCORE) to assess whether SBT has a protective effect against DOX-induced intestinal damage. From the histopathological results of H&E staining, the DOX group exhibited significant tissue damage (marked by arrows), including cell necrosis and structural abnormalities (Figure 6A(a)). In contrast, the SBT + DOX group showed a noticeable reduction in pathological lesions, with better-preserved tissue integrity, suggesting a protective effect of SBT. Furthermore, according to the IHC staining results, SOD2 is an antioxidant enzyme that can reduce oxidative stress-induced damage. The DOX group demonstrated a significant decline in SOD2 expression, whereas the SBT + DOX group showed partial recovery of SOD2 levels (Figure 6B(a), DOX vs. SBT + DOX, *p* < 0.05). This indicates that SBT may reduce DOX-induced oxidative stress via an antioxidant mechanism. TNF-α is an inflammatory marker, and its expression was significantly elevated in the DOX group, while the SBT + DOX group showed a decrease in TNF-α levels (Figure 6B(b), DOX vs. SBT + DOX, *p* < 0.01), suggesting that SBT may help attenuate DOX-induced inflammation.

We quantitatively compared pathological changes in colonic tissue using the tissue lesion score (SCORE) to assess whether SBT has a protective effect against DOX-induced colonic damage. Figure 7 evaluates whether SBT has a protective effect against DOX-induced colonic tissue damage. From the histopathological results of H&E staining, the DOX group exhibited significant tissue damage, whereas the SBT + DOX group showed a noticeable reduction in pathological lesions, suggesting that SBT provides a degree of protection. Furthermore, according to the IHC staining results, SOD2 is an antioxidant enzyme that can reduce oxidative stress-induced damage. The DOX group demonstrated a significant decline in SOD2 expression, whereas the SBT + DOX group showed partial recovery of SOD2 levels (Figure 7B(a), DOX vs. SBT + DOX, *p* < 0.05), indicating that SBT may reduce DOX-induced oxidative stress via an antioxidant mechanism. TNF-α is an inflammatory marker, and its expression was significantly elevated in the DOX group, while the SBT + DOX group showed a decrease in TNF-α levels (Figure 7B(b), DOX vs. SBT + DOX, *p* < 0.05), suggesting that SBT may help attenuate DOX-induced inflammation.

### 2.6. Impact of SBT on Cardiac Dysfunction Induced by DOX Treatment in Mice

We conducted cardiac ultrasound assessments to evaluate the cardiac function of ICR mice subjected to sham, SBT, DOX, and SBT + DOX treatments over a three-week period. Figure 8A(a) presents a B-mode image, which reveals breathing and heartbeat patterns. The results indicate irregular breathing and heartbeat in the DOX group. Figure 8A(b) illustrates an M-mode image, used to monitor variations in cardiac chamber size and systolic and diastolic functions over time. These results demonstrate impaired systolic and diastolic functions along with a decreased rate in the DOX group. Figure 8A(c) shows a PW mode image, which measures blood flow velocity, size, and shape. The findings reveal that DOX chemotherapy may reduce heart rate and blood flow in ICR mice. Figure 8B compares heart rate changes across the four groups, showing that the heart rate in the DOX group was significantly lower than in the sham, SBT, and SBT + DOX groups (Figure 8B, *p* < 0.05). These results suggest that SBT has the potential to mitigate cardiac dysfunction caused by DOX chemotherapy in mice.

We examined the expression of SOD2, an oxidative stress-related protein, and TNF-α, an inflammation-related protein, in the myocardial tissue of ICR mice. Histopathological analysis revealed hemorrhagic damage in the DOX-treated myocardial tissue (Figure 9A(a)). IHC staining results showed that SOD2 expression was significantly lower in the DOX group compared to the SBT and SBT + DOX groups (Figure 9B(a), *p* < 0.01–0.05). Conversely, TNF-α expression was markedly higher in the DOX group relative to the sham, SBT, and SBT + DOX groups (Figure 9B(b), *p* < 0.05). These findings suggest that SBT treatment alleviates DOX-induced myocardial damage in mice by enhancing SOD2 expression and suppressing TNF-α levels, highlighting its potential protective effects against chemotherapy-induced oxidative stress and inflammation.

## 3. Discussion

This study utilized SBT provided by Arjil Pharmaceuticals LLC to investigate its potential in alleviating oxidative stress and inflammation in the gut and heart induced by DOX treatment. The experimental results revealed that feeding DOX-treated mice with SBT for three weeks significantly reduced mortality (Figure 1A), mitigated weight loss (Figure 1B), notably enhanced motor coordination (Figure 2A) and muscle endurance (Figure 2B), alleviated depressive behavior (Figure 3A) while increasing muscle strength (Figure 3B), and reduced sensitivity to cold stimuli (Figure 4). Furthermore, SBT treatment increased the number of fecal granules in DOX-treated mice (Figure 5) and significantly improved cardiac function and heart rate (Figure 7). By enhancing the expression of the oxidative stress-related protein SOD2 and reducing the expression of the inflammation-related protein TNF-α, SBT effectively alleviated the chemotherapy-induced toxicity and damage in the small intestine (Figure 5), colon (Figure 6), and myocardial tissues (Figure 8) caused by DOX treatment. Based on these findings, we propose that supplementing SBT in cancer patients undergoing DOX chemotherapy could effectively alleviate the adverse side effects on the gut and heart, thereby improving the quality of life for patients after chemotherapy.

Oral administration of probiotics to enrich the gut microbiota has been used to alleviate the adverse physical symptoms experienced by patients after chemotherapy and to reduce gastrointestinal side effects, such as diarrhea. Generally, the use of probiotics in clinical settings is known to offer extensive benefits, including alleviation of antibiotic-associated diarrhea and improvement in respiratory tract infections [11]. Currently, several studies are exploring the therapeutic effects of modifying the gut microbiota by administering probiotics as dietary supplements to cancer patients undergoing various treatments. These ongoing studies indicate the immense therapeutic potential of probiotics. A randomized, double-blind clinical trial in cancer patients undergoing colorectal resection found that probiotic administration benefited the composition of gut microbiota and modulated intestinal immune function [12]. Furthermore, a clinical study investigated the safety and efficacy of a probiotic formulation containing multiple bacterial strains in colorectal cancer patients receiving chemotherapy. The study demonstrated a reduced overall incidence of diarrhea in patients treated with probiotics [13]. Additionally, a randomized clinical study involving colorectal cancer patients undergoing colectomy reported significant downregulation of inflammatory and anti-inflammatory cytokines in the intestinal mucosa after treatment with yeast-based probiotics [14].

Probiotics are considered important modulators of gastrointestinal health, altering the gut microbiota by promoting the colonization of beneficial bacteria. While the beneficial effects of probiotics are now widely recognized, emerging evidence indicates that changes in gut microbiota can also influence various other organ systems, including the heart, through a process commonly referred to as the gut-heart axis. Furthermore, cardiac dysfunctions such as heart failure can lead to gut microbiota imbalances, which in turn further exacerbate cardiac dysfunction [15,16]. In terms of the cardiovascular system, the administration of probiotics has been shown to lower blood pressure in both experimental animals and human subjects. For instance, administering a combination of probiotic-fermented *Lactobacillus fermentum* or *Lactobacillus rhamnosus* and *Lactobacillus gasseri* to spontaneously hypertensive mice via drinking water for five weeks significantly reduced systolic blood pressure. This effect was associated with improved aortic endothelium-dependent relaxation responses [17]. A meta-analysis of clinical studies involving 846 hypertensive patients also demonstrated that probiotic treatment significantly reduced both systolic and diastolic blood pressure in hypertensive subjects [18]. The significance of the gut-heart axis is further evident in experimental animals, where antibiotic treatment led to gut microbiota depletion and myocardial infarction in mice, while probiotic treatment enriched gut microbiota and alleviated myocardial infarction [19]. Trimethylamine N-oxide (TMAO) has been proposed as a key link between gut microbiota and cardiovascular diseases, including atherosclerosis and hypertension. However, current data on TMAO-mediated procardiac remodeling effects are lacking, necessitating further studies to elucidate its role in mediating the gut-heart axis. Recently, it has been reported that synbiotic supplementation can reduce chemotherapy-induced complications in patients with gastrointestinal cancer and regulate the number of gut microbiotas to balance the intestinal microecology of the body [20].

This study focuses on the effects of SBT in mitigating symptoms induced by cancer chemotherapy. In our laboratory’s ICR mouse model, the administration of SBT resulted in a significant reduction in colonic mucosal damage and myocardial inflammation. These findings are consistent with previous research utilizing SBT in mouse models of inflammatory bowel disease, where SBT significantly reduced colonic mucosal damage and inflammation. A clinical study also revealed that synbiotic treatment reduced the risk of postoperative complications, such as irritable bowel syndrome, in cancer patients undergoing colorectal cancer resection [21]. Cancer patients undergoing chemotherapy are often susceptible to infections that can lead to complications such as sepsis, heart failure, or gastrointestinal disruption, potentially resulting in hospitalization, interruption of chemotherapy, and accelerated mortality. While this study investigated the use of SBT to counteract symptoms induced by cancer chemotherapy and reduce its side effects, SBT have not yet been approved for clinical use or as part of chemotherapy management protocols. This limitation arises due to insufficient trial efficacy or results that fail to meet clinical or statistical significance. Maintaining proper intestinal function is essential for fostering a healthy gut microbiota. Chemotherapy, however, severely damages the intestinal mucosal layer, leading to the loss of beneficial gut microbes. Current research aims to develop safe approaches to restore intestinal mucosal integrity and reduce dysbiosis, which could mitigate chemotherapy-induced mucosal injury and myocardial inflammation. Furthermore, the efficacy of probiotics can be enhanced by co-administration with prebiotics, forming a synbiotic combination that improves probiotic survival in the gut environment. Future research should focus on well-designed human trials to explore the therapeutic efficacy of SBT in patients undergoing standard cancer chemotherapy.

The role of immunotherapy in cancer treatment is becoming increasingly significant, especially as immune checkpoint inhibitors (such as PD-1/PD-L1 inhibitors) have become standard treatment options for many types of cancer [22]. Compared to traditional chemotherapy, immunotherapy harnesses the patient’s own immune system to attack tumor cells, reducing damage to healthy cells. However, this type of therapy can also trigger immune-related side effects. Research suggests that synbiotics may help reduce chronic inflammation and enhance immune function in patients with metabolic disorders. However, the interaction between immunotherapy and synbiotics remains under investigation. Some studies indicate that changes in gut microbiota could affect the efficacy of immunotherapy, though the exact mechanisms are not yet fully understood. Currently, there are no clear clinical guidelines recommending the use of synbiotic supplements for cancer patients undergoing immunotherapy. However, considering the impact of gut microbiota on immune system function, future research may explore the potential benefits and risks of this combination in greater detail.

## 4. Materials and Methods

### 4.1. SBT Supplement Preparation

In this study, we selected SBT provided from Arjil Pharmaceuticals LLC to examine their effectiveness in relieving DOX-induced heart and gut inflammation. SBT were administered orally after dissolving in water. The SBT supplements, which combine probiotics and prebiotics, aim to improve gut microbiota and may influence immune system regulation. This SBT supplement is a functional dietary supplement designed to support digestive health and overall wellness. It contains a blend of various oligosaccharides, including isomaltooligosaccharide, fructooligosaccharide, lactose sucrose, galactooligosaccharide, xylooligosaccharide, and beetoligosaccharide, which serve as prebiotics to nourish beneficial gut bacteria and enhance gastrointestinal function. The formula is enhanced with probiotics, offering a total concentration of 7.4 × 10⁸ CFU per gram, including a mix of various Lactobacillus (*Lactobacillus acidophilus* X1, *Lactobacillus casei* X1, *Lactobacillus plantarum* X1, *Lactobacillus rhamnosus* X2, *Lactobacillus paracasei* X2, *Lactobacillus reuteri* X2, *Lactobacillus gasseri* X1, *Lactobacillus fermentum* X1, *Lactobacillus helveticus* X1, *Lactobacillus johnsonii* X1, *Lactobacillus salivarius* X1) and Bifidobacterium strains (*Bifidobacterium animalis subsp. lactis* X2, *Bifidobacterium longum subsp. Infantis* X1, *Bifidobacterium breve* X1, *Bifidobacterium bifidum* X2), which contribute to a balanced gut microbiome. Inulin from chicory fiber provides additional prebiotic support, improving digestive function. The supplement also contains pineapple enzyme, which aids in digestion by breaking down proteins and electrolytes such as sodium chloride and sodium bicarbonate.

### 4.2. Experimental Design

In this experiment, we used 24 male ICR mice, which were purchased from the National Laboratory Animal Center (NLAC, Taipei, Taiwan), divided into the sham group (N = 6), SBT treatment group (SBT, N = 6), DOX treatment group (DOX, N = 6), and SBT + DOX treatment group (SBT + DOX group, N = 6). The animal experiment lasted a total of five weeks. DOX is a chemotherapy drug used to treat various types of cancer, often causing adverse side effects such as gastrointestinal inflammation, nausea, vomiting, acute myocardial infarction, and myocarditis in chemotherapy patients. The duration of this study was chosen based on previous models assessing DOX-induced toxicity [23,24]. The experimental design flowchart of this study is shown in Figure 10. In total, 24 male ICR mice (18.5 ± 0.4 g) were acclimated in the animal housing facility for one week. During the experimental period, 6 male ICR mice with sham treatment comprised the control group that received regular feeding and water for five consecutive weeks; 6 male ICR mice with SBT treatment comprised the positive control group that received regular feeding and water for five weeks with SBT supplementation twice daily in the morning and evening at a dose of 1.5 mg/kg; 6 male ICR mice with DOX treatment comprised the experimental control group that received regular feeding and water for five weeks with intraperitoneal injections of DOX every other day at a dose of 2.4 mg/kg for five weeks; and 6 male ICR mice with SBT + DOX treatment comprised the experimental group that received regular feeding and water for five weeks with SBT supplementation twice daily (1.5 mg/kg per dose) in addition to DOX injections every other day (2.4 mg/kg per dose) for five weeks. During the experiment, survival rates and body weight changes were recorded daily. Behavioral tests were conducted weekly, including a rotating rod test, treadmill running test, forced swim test, grip strength test, and cold sensitivity test. Due to mortality in DOX-treated mice starting from the fourth week, behavioral test data were averaged and analyzed using the results from the third week. At the end of the fifth week, perfusion was performed, and intestinal and cardiac tissues were collected. These tissues were sectioned for H&E and IHC staining. The animal experiment complied with the NIH Guidelines for the Care and Use of Laboratory Animals and was approved by the NTNU Institutional Animal Care and Use Committee (Protocol Number: NTNU Animal Experiment Approval No. 110017).

### 4.3. Forced Swim Test

The forced swim test (FST) is a well-established model commonly utilized to evaluate depressive-like behaviors in rodents. In this test, young mice are placed in a transparent cylinder, an aversive and confined environment, where they instinctively struggle to escape. Pretreatment with antidepressants has been shown to reduce overall immobility and increase the latency to the first immobility event. For the procedure, each mouse is introduced into a glass beaker (15 cm in diameter, 20 cm in height) filled with tap water maintained at a temperature of 20–25 °C, reaching a depth of 16 cm. The water level must be sufficient to prevent the mouse from touching the bottom of the container with its hind limbs or tail and should be adjusted to accommodate variations in the mice’s size. The test duration is 6 min, after which the mice are carefully dried and returned to their home cages. An independent observer records the session and quantifies the duration of immobility. Immobility, or “floating”, is defined as the minimal movement necessary to keep the mouse’s head above the water’s surface. Although the forced swim test is utilized less frequently than the light/dark box (LDB) test, prior studies have reported that cranial irradiation increases immobility time, which is interpreted as an indicator of enhanced depressive-like behavior.

### 4.4. Cold Sensitivity Score

The Cold Plantar Assay (CPA) is a behavioral testing method commonly used to evaluate cold sensitivity in mice. The procedure involves placing the mice on a transparent glass plate and allowing them to acclimate to their surroundings. Small pellets of compressed dry ice are then applied to the hind paws of the mice through the glass plate. The latency of paw withdrawal in response to the cold stimulus is recorded as an indicator of cold sensitivity.

### 4.5. Cardiac Ultrasound Scan

Echocardiography (S-Sharp Corporation, Taichung, Taiwan) employs ultrasonic wave reflection to measure object size and distance. This noninvasive, nonradiative technique provides real-time dynamic imaging and allows for repeated assessments, making it a highly effective method for evaluating cardiac conditions. An ultrasound probe emits sound waves that penetrate the chest wall and reflect upon contacting the heart. The probe captures these returning waves, with their travel time indicating the distance between the probe and the reflective surface. The intensity of the reflected waves reveals the properties of the interface, resulting in a detailed cardiac image. This enables the assessment of various cardiac parameters, including ventricular size, myocardial thickness, and systolic and diastolic function as well as the detection of valve stenosis or regurgitation. Additionally, echocardiography can evaluate blood flow direction and velocity within cardiac vessels.

### 4.6. Hematoxylin and Eosin (H&E) Staining

This study mainly used H&E (hematoxylin and eosin, Sigma-Aldrich Co., St. Louis, MO, USA) staining to examine and analyze myocardial and intestinal tissue sections of ICR mice, allowing differentiation of various cell types and tissue structures under a microscope. Deparaffinized tissue sections are sequentially immersed in three cylinders of 100% alcohol for 1 min each to remove residual xylene. Subsequently, rehydration is performed by immersing the sections in 95% alcohol for 1 min, 75% alcohol for 1 min, and deionized water for 3 min, completing the hydration process. For nuclear staining, the sections are immersed in hematoxylin solution for 7 min to stain basophilic structures. The slides are then carefully rinsed in a water bath for 1 min with tap water to remove excess hematoxylin. They are subsequently immersed in differentiation solution for 3–5 s to remove stain from the cytoplasm, followed by another careful rinse with tap water for 1 min. Afterward, the sections are immersed in 75% alcohol for 1 min and 95% alcohol for 1 min to complete the washing process. Next, for cytoplasmic staining, the sections are immersed in an eosin solution for 40 s. Dehydration is then performed by sequential immersion in two cylinders of 100% alcohol for 1 min and 3 min, respectively. Finally, the slides undergo clearing by immersion in two cylinders of xylene for 1 min and 3 min, respectively. The H&E staining procedure is completed at this stage. For mounting, mounting medium is applied to the tissue section and a coverslip is placed, ensuring no air bubbles are trapped. Once the mounting medium has solidified, the slides are ready for high-power microscopic imaging.

### 4.7. Immunohistochemical (IHC) Staining

This study mainly used IHC staining to examine and analyze the expression of superoxide dismutase 2 (SOD2) and tumor necrosis factor-α (TNF-α) in the myocardium and intestinal tissues of ICR mice. In this study, myocardial and intestinal tissue sections were stained immunohistochemically with SOD2 and TNF-α antibodies (Cell Signaling Technology Inc., Danvers, MA, USA; 1 h at room temperature). IHC staining is a complex procedure comprising several critical steps, including fixation, microtomy, pretreatment, blocking, primary antibody application, detection systems, and chromogen reagents. Tissue sections are typically baked at 60 °C for 2–4 h; however, insufficient drying can weaken tissue-to-slide adhesion, leading to damage or detachment during heat pretreatment. To ensure optimal adherence and minimize detachment, overnight baking at 60 °C is recommended. The deparaffinization and rehydration protocols vary depending on the reagents employed. Deparaffinization is often carried out using xylene or its substitutes, while rehydration utilizes graded alcohols. Incomplete deparaffinization may hinder antibody recognition of tissue antigens, leading to weak or absent staining. Residual paraffin can also result in background staining. Replacing xylene and alcohol regularly is advised to prevent oversaturation that could compromise deparaffinization and staining quality. Antigen retrieval is a crucial step involving Heat-Induced Epitope Retrieval (HIER) or Enzyme-Induced Epitope Retrieval (EIER). Reagent selection and procedural accuracy are key to successful retrieval. Incomplete antigen retrieval or the use of inappropriate reagents can impair antigen–antibody binding, causing weak or absent staining. Conversely, excessive retrieval may lead to nonspecific binding, resulting in background staining. Commonly used antigen retrieval reagents include citrate buffer (pH 5–6) and EDTA buffer (pH 8–9). Pressure cookers are favored for their stable temperature control and timing, contributing to standardized IHC outcomes. For most antibodies, HIER with EDTA buffer (pH 8–9) yields optimal results. Innovative reagents such as Trilogy and Declere streamline the IHC pretreatment process by combining deparaffinization, rehydration, and antigen retrieval into a single workflow. These reagents enhance consistency and reproducibility in formalin-fixed, paraffin-embedded tissue section staining. To determine the relative expression levels of SOD2 and TNF-α, ImageJ 1.53 (National Institutes of Health, Bethesda, USA) was used for IHC quantification. The sham group was designated as the reference control, with its expression levels normalized to 100%, allowing comparative analysis across experimental groups. The histopathological SCORE, which demonstrates the protective effect of treatment with SBT and the reduction of lesions caused by DOX, is as follows: First, ImageJ was used for IHC analysis to enhance quantitative evaluation accuracy. The IHC Profiler plugin was installed, and high-resolution IHC images were loaded. Color deconvolution was applied to separate the target staining, followed by thresholding to identify positive regions. Next, grayscale analysis was used to determine staining intensity, and the proportion of positive cells was measured. Finally, the IHC Score was calculated based on staining intensity and positive cell percentage, providing a standardized assessment of protein expression.

### 4.8. Statistical Analysis

The results are expressed as the mean ± standard error of the mean (SEM) for each group. Statistical significance was assessed using one-way analysis of variance (ANOVA), followed by post hoc comparisons with the Student–Newman–Keuls (S-N-K) multiple range test. A *p*-value of <0.05 was considered statistically significant for identifying differences between groups. We used GraphPad Prism 6 (GraphPad Prism 6.0 Software Inc., San Diego, CA, USA) for graph preparation. All statistical analyses were performed using IBM SPSS Statistics software 29.0 (Armonk, NY, USA).

## 5. Conclusions

Our study explored the protective effects of SBT on chemotherapy-induced intestinal and cardiac inflammation in animal models. The findings revealed that SBT significantly mitigates DOX-induced cardio-intestinal damage, enhancing heart function and reducing intestinal inflammation. Furthermore, SBT improved overall health parameters, including behavioral performance, which positively influenced chemotherapy-induced behavioral disturbances in mice. Based on these results, SBT shows potential as an adjunct to chemotherapy; however, further well-designed human trials are necessary to evaluate its therapeutic efficacy in cancer patients undergoing standard chemotherapy.

## 6. Patents

The findings presented in this manuscript are currently under patent application in Taiwan.

## Figures and Tables

**Figure 1 ijms-26-05136-f001:**
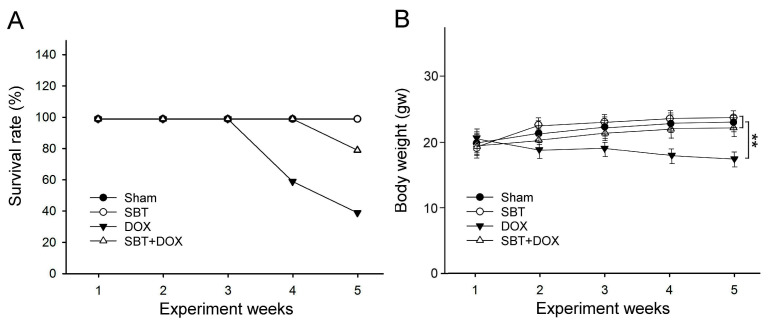
(**A**) Survival rates and (**B**) changes in body weight among ICR mice subjected to sham, SBT, DOX, and SBT + DOX treatments over three consecutive weeks. (N = 6 per group, (**A**): Kaplan–Meier test for the survival evaluation; (**B**): values are expressed as mean ± SEM, ** *p* < 0.01, one-way ANOVA followed by Student–Newman–Keuls test for multiple comparisons).

**Figure 2 ijms-26-05136-f002:**
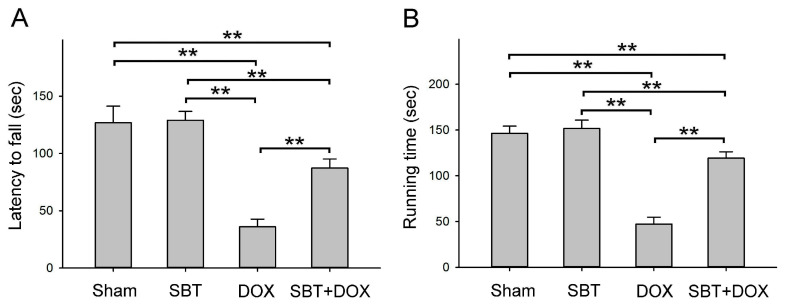
(**A**) Latency to fall time on rotarod, and (**B**) running time on the treadmill among ICR mice subjected to sham, SBT, DOX, and SBT + DOX treatments over three consecutive weeks. (Values are expressed as mean ± SEM, number of samples per group = 6, ** *p* < 0.01, one-way ANOVA followed by Student–Newman–Keuls test for multiple comparisons).

**Figure 3 ijms-26-05136-f003:**
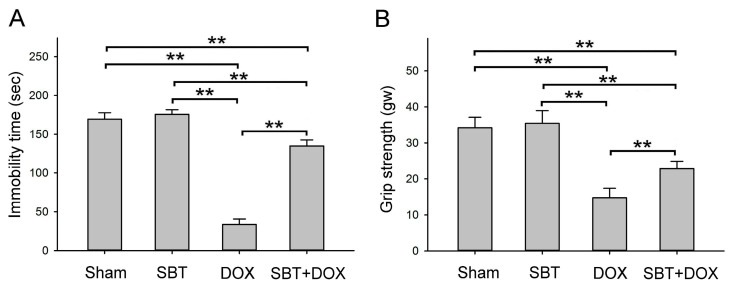
(**A**) Immobility time of forced swimming and (**B**) grip strength among ICR mice subjected to sham, SBT, DOX, and SBT + DOX treatments over three consecutive weeks. (Values are expressed as mean ± SEM, number of samples per group = 6, ** *p* < 0.01, one-way ANOVA followed by Student–Newman–Keuls test for multiple comparisons).

**Figure 4 ijms-26-05136-f004:**
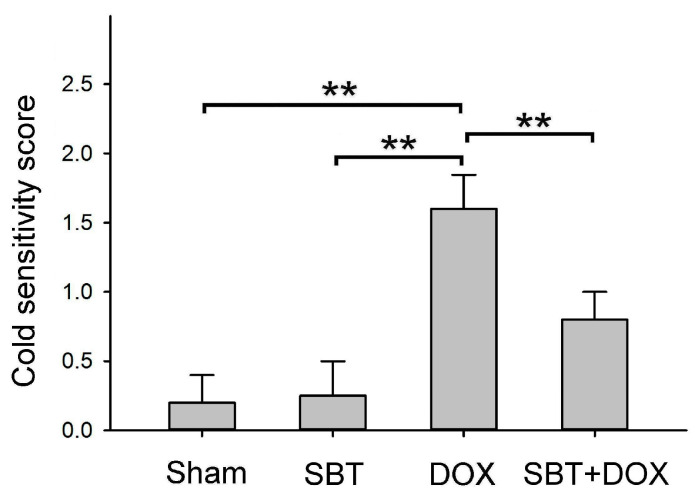
Cold sensitivity score among ICR mice subjected to sham, SBT, DOX, and SBT + DOX treatments over three consecutive weeks. (Values are expressed as mean ± SEM, number of samples per group = 6, ** *p* < 0.01, one-way ANOVA followed by Student–Newman–Keuls test for multiple comparisons).

**Figure 5 ijms-26-05136-f005:**
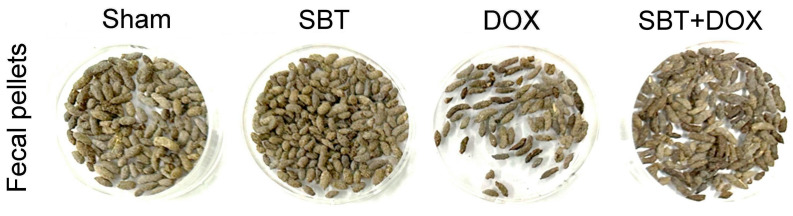
Fecal pellet number among ICR mice subjected to sham, SBT, DOX, and SBT + DOX treatments over three consecutive weeks.

**Figure 6 ijms-26-05136-f006:**
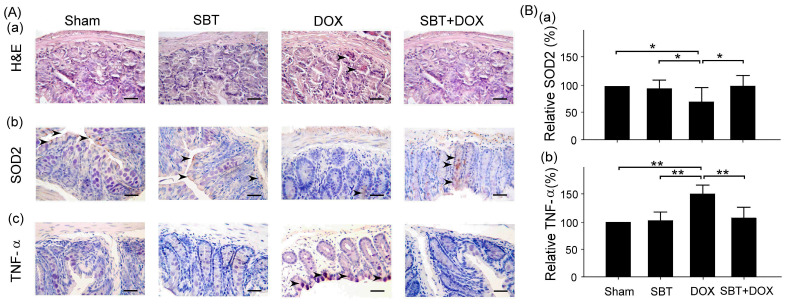
(**A**) Hematoxylin and eosin (H&E) staining (**Aa**) of small intestinal tissues and immunohistochemistry (IHC) analysis of the antioxidant stress-related protein SOD2 (**Ba**) and the inflammation-related protein TNF-α (**Bb**); (**B**) relative expression levels of SOD2 (**Ba**) and TNF-α (**Bb**) in small intestinal tissues of ICR mice subjected to sham, SBT, DOX, and SBT + DOX treatments over five consecutive weeks. Arrows indicate hemorrhagic lesions (**Aa**), SOD2 expression (**Ab**), and TNF-α expression (**Ac**) in the small intestines. Scale bar = 100 μm. Data are presented as mean ± SEM (n = 3 per group), with ** *p* < 0.01 and * *p* < 0.05 determined by one-way ANOVA followed by the Student–Newman–Keuls test for multiple comparisons.

**Figure 7 ijms-26-05136-f007:**
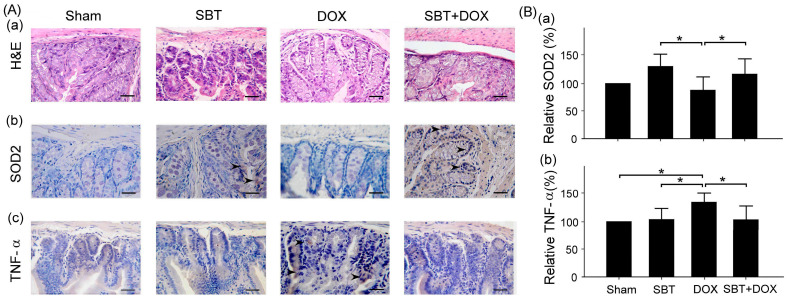
(**A**) H&E (**Aa**) of colon tissue and IHC staining of antioxidant stress-related protein SOD2 (**Ab**) and inflammation-related protein TNF-α (**Ac**); (**B**) comparison of relative expressions of SOD2 (**Ba**) and TNF-α (**Bb**) of colon tissue among ICR mice subjected to sham, SBT, DOX, and SBT + DOX treatments over five consecutive weeks.The arrows indicate the expressions of SOD2 (**Ab**) and TNF-α (**Ac**) in the colon tissue. Scale bar = 100 μm. (Values are expressed as mean ± SEM, number of samples per group = 3, * *p* < 0.05, one-way ANOVA followed by Student–Newman–Keuls test for multiple comparisons).

**Figure 8 ijms-26-05136-f008:**
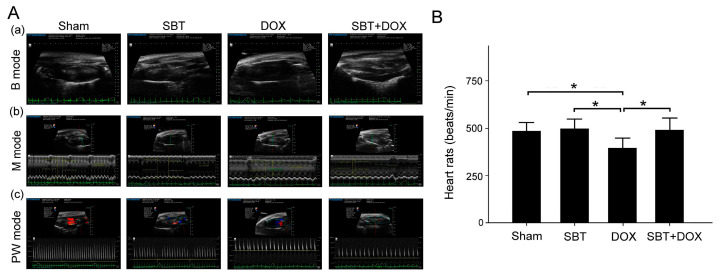
(**A**) Cardiac ultrasound scan of ICR mice in (**a**) B mode; (**b**) M mode; and (**c**) PW mode; (**B**) heart rate among ICR mice subjected to sham, SBT, DOX, and SBT + DOX treatments over three consecutive weeks. (Values are expressed as mean ± SEM, number of samples per group = 6, * *p* < 0.05, one-way ANOVA followed by Student–Newman–Keuls test for multiple comparisons).

**Figure 9 ijms-26-05136-f009:**
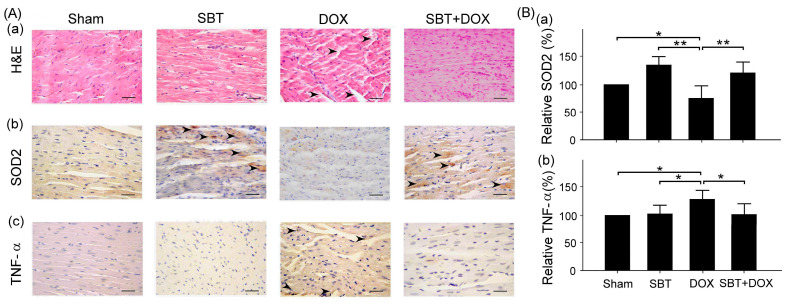
(**A**) H&E (**Aa**) of myocardial tissue and IHC staining of antioxidant stress-related protein SOD2 (**Ab**) and inflammation-related protein TNF-α (**Ac**); (**B**) comparison of relative expressions of SOD2 (**Ba**) and TNF-α (**Bb**) of myocardial tissue among ICR mice subjected to sham, SBT, DOX, and SBT + DOX treatments over five consecutive weeks. The arrows indicate the expressions of hemorrhagic lesions (**Aa**), SOD2 (**Ab**), and TNF-α (**Ac**) in the myocardial tissue. Scale bar = 100 μm. (Values are expressed as mean ± SEM, number of samples per group = 3, ** *p* < 0.01, * *p* < 0.05, one-way ANOVA followed by Student–Newman–Keuls test for multiple comparisons).

**Figure 10 ijms-26-05136-f010:**
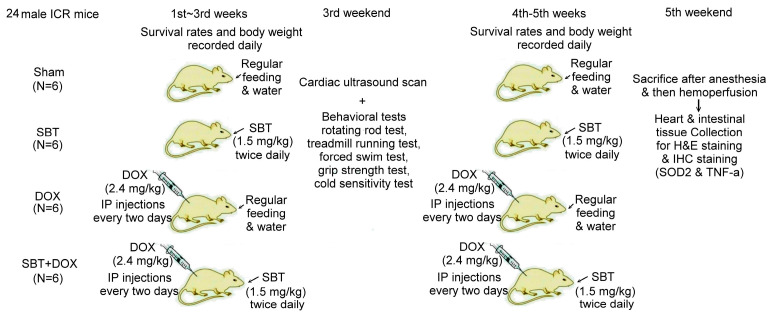
The experimental design flowchart illustrates the experiment assessing the impact of SBT on DOX-induced damage.

## Data Availability

All data are presented in the article.

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
