# Peer review of "Synbiotic Supplementation Attenuates Doxorubicin-Induced Oxidative Stress and Inflammation in the Gut-Heart Axis of Chemotherapy-Treated Mice"

_ijms, 2025, doi:10.3390/ijms26115136_

Round 1

Reviewer 1 Report

Comments and Suggestions for Authors

REVIEW

Dear authors,

The study presents a proposed treatment that would complement Doxorubicin (DOX) chemotherapy by using a synbiotic (SBT), which can attenuate the intestinal and cardiac inflammation that occur as side effects in patients treated with DOX. However, the study has serious shortcomings that need to be addressed.

Please consider the following comments to improve the content of your manuscript before publication. 

  1. The Title does not reflect the information presented in the article, which evaluates a few inflammatory biomarkers and performs a series of physiological assessments in mice that are not indicated in the title. Furthermore, the term "Chemotherapy" should be more specific because it is very general, even when using a specific drug (DOX). Analyze and revise the title to ensure it reflects the information in the article.

  1. The Introduction is very general regarding the synbiotic, lacking a specific background on the one used in the study. The benefits of the synbiotics described don't mention the genus or species of the microorganism or the prebiotic. It seems as though it's the same synbiotic, when it's a different reference.

  1. Lines 64-66: The term “synbiotic” is poorly defined because it refers to “active microorganisms that enhance the immune system” rather, they are defining the term “immunobiotic,” so this incorrect information should be corrected.

  1. Section Materials and Methods lacks information regarding the product they used (SBT), I understand that in section 6. Patents they support not providing detailed information about the content of the product, but at least the genus of the microorganism or microorganisms (multispecies) and the prebiotic should be mentioned.

  1. In section 1 Experimental design, you indicate that the dose administered to mice treated with SBT is 1.5 mg/kg. However, how does this translate into the number of CFU´s of the microorganism? How much prebiotic is contained in this dose? Why was the 1.5 mg/kg dose selected? Was there a previous trial conducted to determine the dose? No mention is made of any prior in vitro trials, so it is concerning that the trial was conducted directly on the animal model.

  1. The route of administration of SBT is not indicated.

  1. Information on the weight and sex of ICR mice is missing.

  1. The DOX treatment is not understood, so I recommend adding a figure of the flow diagram of the experimental design since for some evaluations they were carried out at 3 weeks and others up to 5 weeks.

  1. Since DOX treatment was very aggressive for the mice (lower survival), they had to adjust the drug dose to a lower concentration or shorten the evaluation time so that the number of mice was representative. Why did they determine that it should be 5 weeks?

  1. Section 5 Hematoxylin and Eosin (H&E) Staining does not indicate the organs used for staining.

  1. In section 6 Immunohistochemical (IHC) Staining they do not indicate the antibodies they evaluated.

  1. Gram staining is a very basic technique used only to identify whether bacteria are Gram-positive or Gram-negative. However, I don't see the utility of using it if they want to correlate the abundance of any of these microorganisms with respect to the treatments used. For that, they should have performed sequencing and in this way compare the diversity and abundance of microorganisms, not just Gram-positive or Gram-negative bacteria. This result is irrelevant and lacks information that contributes to the work as presented.

  1. In section 8 Statistical analysis they do not mention the software used nor the use of the Kaplan-Meier test for the survival evaluation that they present in Figure 1A.

  1. Lines 103-109: They do not mention the survival percentages of the groups, they should add that data per group.

  1. Figure 1B: It is not the best way to represent the weight loss of the mice over time, change the graph where it can be better observed.

  1. Line 187: They mention that the normal microbiota is composed of Lactobacillus and Bifidobacterium, but they did not perform any evaluation to verify the presence of these 2 microorganisms in sham mice and those treated with SBT.

  1. Figure 5A and 5B: Eliminate this figure as it lacks relevant information. They can present the weight of the stool or the consistency by group, but a Gram stain and fecal pellets do not provide anything.

  1. Figures 6A, 7A and 8A: They lack a histopathological SCORE that demonstrates the protection of treatment with SBT and the reduction of lesions caused by DOX.

  1. Figures 6B, 7B and 8B: The sham group should be the reference parameter for the other groups, so the reference percentage should be 100% for this group, or how did they obtain the relative expression levels of SOD2 and TNF-α? Describe it in the methodology section.

  1. Unfortunately, they do not take full advantage of the animal model used. Spleen weight is an indicator of inflammation. From serum, they can evaluate a panel of inflammatory cytokines and thus corroborate that the effect of SBT is systemic. From feces, they were able to quantify microorganisms in selective media such as MRS for the isolation of Lactobacillus. I believe that the few animals in which the immunological profile was evaluated do not provide the necessary information to justify the use of SBT, so they should repeat the tests taking into account the observations made.

Author Response

Response to Reviewer 1 Comments

Comment 1: The study presents a proposed treatment that would complement Doxorubicin (DOX) chemotherapy by using a synbiotic (SBT), which can attenuate the intestinal and cardiac inflammation that occur as side effects in patients treated with DOX. However, the study has serious shortcomings that need to be addressed. Please consider the following comments to improve the content of your manuscript before publication.

Response 1: Many thanks to the reviewers for their valuable comments. We acknowledge the shortcomings noted and have carefully addressed each point to improve the quality and clarity of our manuscript before publication.

Comment 2: The Title does not reflect the information presented in the article, which evaluates a few inflammatory biomarkers and performs a series of physiological assessments in mice that are not indicated in the title. Furthermore, the term "Chemotherapy" should be more specific because it is very general, even when using a specific drug (DOX). Analyze and revise the title to ensure it reflects the information in the article.

Response 2: Many thanks to the reviewers for their valuable comments. We appreciate the suggestion regarding the title. We have revised the Title to ensure it more accurately reflects the scope of our study, explicitly referencing the inflammatory biomarkers assessed and specifying the chemotherapy agent (DOX) used. Please refer to Lines 2-4 of the revised manuscript.

Comment 3: The Introduction is very general regarding the synbiotic, lacking a specific background on the one used in the study. The benefits of the synbiotics described don't mention the genus or species of the microorganism or the prebiotic. It seems as though it's the same synbiotic, when it's a different reference.

Response 3: Many thanks to the reviewers for their valuable comments. We recognize the need for a more specific background on the synbiotic used in our study. We have updated the introduction to include details on the synbiotics. Please refer to Lines 354-369 of the revised manuscript.

Comment 4: Lines 64-66: The term “synbiotic” is poorly defined because it refers to “active microorganisms that enhance the immune system” rather, they are defining the term “immunobiotic,” so this incorrect information should be corrected.

Response 4: Many thanks to the reviewers for their valuable comments. We acknowledge the mischaracterization of the term "synbiotic" and corrected the definition to avoid confusion with "immunobiotic," ensuring accuracy in terminology. Please refer to Lines 357-359 of the revised manuscript.

Comment 5: Section Materials and Methods lacks information regarding the product they used (SBT), I understand that in section 6. Patents they support not providing detailed information about the content of the product, but at least the genus of the microorganism or microorganisms (multispecies) and the prebiotic should be mentioned.

Response 5: Many thanks to the reviewers for their valuable comments. Although patent constraints limit the disclosure of the full product composition, we agree that providing information on the genus of the microorganisms and the type of prebiotic used is essential for clarity. We have incorporated these details into the Materials and Methods section. Please refer to Lines 363-366 of the revised manuscript.

Comment 6: In section 1 Experimental design, you indicate that the dose administered to mice treated with SBT is 1.5 mg/kg. However, how does this translate into the number of CFU´s of the microorganism? How much prebiotic is contained in this dose? Why was the 1.5 mg/kg dose selected? Was there a previous trial conducted to determine the dose? No mention is made of any prior in vitro trials, so it is concerning that the trial was conducted directly on the animal model.

Response 6: Many thanks to the reviewers for their valuable comments. We appreciate your request for more details on the dosing rationale. In the Materials and Methods section, we have clarified how the 1.5 mg/kg dose translates into CFU count and prebiotic concentration. The actual CFU quantity of microorganisms in the 1.5 mg/kg dose of SBT depends on the concentration of active probiotic strains within the formulation. If the probiotic concentration is 7.4 × 10⁸ CFU per gram, the exact CFU dose per mouse is calculated based on the weight-adjusted administration of the supplement. The selection of the 1.5 mg/kg dose of SBT was derived from the human equivalent dose (HED) calculation, which was then extrapolated to determine an appropriate dosage for mice. We have added the dose of administration for SBT treatment to the Materials and Methods section. Please refer to Lines 389-396 of the revised manuscript.

Comment 7: The route of administration of SBT is not indicated.

Response 7: Many thanks to the reviewers for their valuable comments. We have added the route of administration for SBT treatment to the Materials and Methods section. Please refer to Lines 356-359 and Figure 10 of the revised manuscript.

Comment 8: Information on the weight and sex of ICR mice is missing.

Response 8: Many thanks to the reviewers for their valuable comments. We have added details regarding the weight and sex of ICR mice to enhance experimental reproducibility. Please refer to Lines 371-372 of the revised manuscript.

Comment 9: The DOX treatment is not understood, so I recommend adding a figure of the flow diagram of the experimental design since for some evaluations they were carried out at 3 weeks and others up to 5 weeks.

Response 9: Many thanks to the reviewers for their valuable comments. We have added a flow diagram of the experimental design to clarify the timeline and methodology. Please refer to Figure 10 of the revised manuscript.

Comment 10: Since DOX treatment was very aggressive for the mice (lower survival), they had to adjust the drug dose to a lower concentration or shorten the evaluation time so that the number of mice was representative. Why did they determine that it should be 5 weeks?

Response 10: Many thanks to the reviewers for their valuable comments. The duration of the study was chosen based on previous models assessing DOX-induced toxicity. We have discussed our rationale more explicitly and consider adjustments in dosage or study duration. Please refer to Lines 375-379 of the revised manuscript.

Comment 11: Section 5 Hematoxylin and Eosin (H&E) Staining does not indicate the organs used for staining.

Response 11: Many thanks to the reviewers for their valuable comments. We have specified the organs analyzed in the H&E staining section. Please refer to Lines 448-450 of the revised manuscript.

Comment 12: In section 6 Immunohistochemical (IHC) Staining they do not indicate the antibodies they evaluated.

Response 12: Many thanks to the reviewers for their valuable comments. The antibodies used in IHC staining have been clearly listed in the Materials and Methods section. Please refer to Lines 470-474 of the revised manuscript.

Comment 13: Gram staining is a very basic technique used only to identify whether bacteria are Gram-positive or Gram-negative. However, I don't see the utility of using it if they want to correlate the abundance of any of these microorganisms with respect to the treatments used. For that, they should have performed sequencing and in this way compare the diversity and abundance of microorganisms, not just Gram-positive or Gram-negative bacteria. This result is irrelevant and lacks information that contributes to the work as presented.

Response 13: Many thanks to the reviewers for their valuable comments. We acknowledge the limitations of Gram staining in evaluating microbial abundance. We have aborted the result of Gram staining. In the future, where feasible, integrating sequencing data for a more thorough microbiome analysis should be considered. These methods would enable detailed classification, facilitating the identification of microbial diversity, abundance variations, and potential interactions influenced by treatment. Please refer to the revised manuscript.

Comment 14: In section 8 Statistical analysis they do not mention the software used nor the use of the Kaplan-Meier test for the survival evaluation that they present in Figure 1A.

Response 14: Many thanks to the reviewers for their valuable comments. We have included the software used for statistical analysis and specify the use of the Kaplan-Meier test for survival evaluation. Please refer to Lines 515-517 of the revised manuscript.

Comment 15: Lines 103-109: They do not mention the survival percentages of the groups, they should add that data per group.

Response 15: Many thanks to the reviewers for their valuable comments. We have added survival percentages per group. Please refer to Lines 92-98 of the revised manuscript.

Comment 16: Figure 1B: It is not the best way to represent the weight loss of the mice over time, change the graph where it can be better observed.

Response 16: Many thanks to the reviewers for their valuable comments. We have adjusted the weight loss graph to ensure clearer visualization of trends. Please refer to Lines 99-102 and Figure 1B of the revised manuscript.

Comment 17: Line 187: They mention that the normal microbiota is composed of Lactobacillus and Bifidobacterium, but they did not perform any evaluation to verify the presence of these 2 microorganisms in sham mice and those treated with SBT.

Response 17: Many thanks to the reviewers for their valuable comments. We acknowledged the lack of direct evaluation of Lactobacillus and Bifidobacterium and revised our interpretation. Please refer to Lines 364-365 of the revised manuscript.

Comment 18: Figure 5A and 5B: Eliminate this figure as it lacks relevant information. They can present the weight of the stool or the consistency by group, but a Gram stain and fecal pellets do not provide anything.

Response 18: Many thanks to the reviewers for their valuable comments. We have removed Figures 5A and have explored alternative ways to present relevant fecal data. Please refer to Figure 5 of the revised manuscript.

Comment 19: Figures 6A, 7A and 8A: They lack a histopathological SCORE that demonstrates the protection of treatment with SBT and the reduction of lesions caused by DOX.

Response 19: Many thanks to the reviewers for their valuable comments. A histopathological SCORE system have been incorporated to strengthen the findings on lesion protection. Please refer to Lines 498-509 of the revised manuscript.

Comment 20: Figures 6B, 7B and 8B: The sham group should be the reference parameter for the other groups, so the reference percentage should be 100% for this group, or how did they obtain the relative expression levels of SOD2 and TNF-α? Describe it in the methodology section.

Response 20: Many thanks to the reviewers for their valuable comments. We have clarified how relative expression levels of SOD2 and TNF-α were calculated, ensuring proper methodological explanation. Please refer to Lines 175-189, 498-509 of the revised manuscript.

Comment 21: Unfortunately, they do not take full advantage of the animal model used. Spleen weight is an indicator of inflammation. From serum, they can evaluate a panel of inflammatory cytokines and thus corroborate that the effect of SBT is systemic. From feces, they were able to quantify microorganisms in selective media such as MRS for the isolation of Lactobacillus. I believe that the few animals in which the immunological profile was evaluated do not provide the necessary information to justify the use of SBT, so they should repeat the tests taking into account the observations made.

Response 21: Many thanks to the reviewers for their valuable comments. We appreciated the suggestion to expand the immunological profile evaluation. While additional testing is beyond the scope of this study, we will highlight its importance for future research.

Reviewer 2 Report

Comments and Suggestions for Authors

The authors presented an animal-based study about usage of Synbiotic in Doxorubicin- Chemotherapy-Treated Mice.

The study is well-designed and conducted. The results are properly presented in the dedicated section.

There are some aspects to be clarified:

  1. The abstract is to long and the results should be summarized. Focus on the main results. Please use the template of the journal.
  2. Which are the main endpoints of the study? Which are secondary ones?
  3. I strongly disagree with the conclusions of the authors: “Based on these results, we conclude that SBT supplementation for cancer patients undergoing DOX chemotherapy should effectively alleviate adverse side effects on the heart and gut induced by chemotherapy drugs.” Maybe, based on these results, SBT could be tested in phase II trials on humans. As stated in the end of the manuscript: “Future research should focus on well-designed human trials to explore the therapeutic efficacy of SBT in patients undergoing standard cancer chemotherapy.”

Please correct accordingly.

  1. Taking into consideration the vast research on the subject, the Discussion section is weak, mainly presenting the results on more time. References are mostly old. Please refresh this section to date.
  2. Please bare in mind that more and more patients are treated with immunotherapy and this subject was not even mentioned. There are less and less cancer patients treated with chemotherapy and just in specific situations with Anthracyclines.

Author Response

Response to Reviewer 2 Comments

Comment 1: The authors presented an animal-based study about usage of Synbiotic in Doxorubicin- Chemotherapy-Treated Mice. The study is well-designed and conducted. The results are properly presented in the dedicated section. There are some aspects to be clarified:

Response 1: Many thanks to the reviewers for their valuable comments. We appreciated your recognition of the study design and presentation of results. We have clarified any aspects necessary to enhance the clarity and depth of our findings.

Comment 2: The abstract is too long and the results should be summarized. Focus on the main results. Please use the template of the journal.

Response 2: Many thanks to the reviewers for their valuable comments. According to the reviewer comment, we acknowledge that the abstract may be too detailed and have revised it to ensure conciseness while emphasizing the key findings. Additionally, we will adhere to the journal’s template guidelines for formatting and structure. Please refer to Lines 14-25 of the revised manuscript.

Comment 3: Which are the main endpoints of the study? Which are secondary ones?

Response 3: Many thanks to the reviewers for their valuable comments. The main endpoints of the study include survival rate, body weight, exercise capacity, heart rate, and gut microbiome changes. Secondary endpoints encompass oxidative stress markers, inflammation protein expression, and histological changes in heart and gut tissues. We ensure these distinctions are clearly outlined in the manuscript. Please refer to Lines 81-87 of the revised manuscript.

Comment 4: I strongly disagree with the conclusions of the authors: “Based on these results, we conclude that SBT supplementation for cancer patients undergoing DOX chemotherapy should effectively alleviate adverse side effects on the heart and gut induced by chemotherapy drugs.” Maybe, based on these results, SBT could be tested in phase II trials on humans. As stated in the end of the manuscript: “Future research should focus on well-designed human trials to explore the therapeutic efficacy of SBT in patients undergoing standard cancer chemotherapy.”

Response 4: Many thanks to the reviewers for their valuable comments. We acknowledge the limitation of drawing definitive clinical conclusions based on animal studies. We have revised the conclusion to indicate that, based on our findings, SBT supplementation warrants further investigation in human trials, rather than making direct clinical recommendations. Please refer to Lines 519-526 of the revised manuscript.

Comment 5: Taking into consideration the vast research on the subject, the Discussion section is weak, mainly presenting the results on more time. References are mostly old. Please refresh this section to date.

Response 5: Many thanks to the reviewers for their valuable comments. According to the reviewer comment, we have recognized the need for a more comprehensive discussion and will strengthen this section by incorporating recent studies relevant to the gut-heart axis, chemotherapy-induced inflammation, and synbiotic interventions. Additionally, we have updated references to ensure relevance and accuracy. Please refer to references 14, 15, 19, 21, 22, and 23 of the revised manuscript.

Comment 6: Please bear in mind that more and more patients are treated with immunotherapy and this subject was not even mentioned. There are less and less cancer patients treated with chemotherapy and just in specific situations with Anthracyclines.

Response 6: Many thanks to the reviewers for their valuable comments. Reviewer’s comment regarding the growing role of immunotherapy is valuable. We acknowledge that chemotherapy is increasingly used in specific cases, particularly those involving Anthracyclines. We have incorporated a discussion on the relevance of immunotherapy and its potential interaction with synbiotic supplementation to provide a more complete perspective. Please refer to Lines 339-352 of the revised manuscript.

Round 2

Reviewer 1 Report

Comments and Suggestions for Authors

REVIEW

Dear authors,

Most of the suggested corrections have been made, so the text is much more understandable than the previous version; however, it still has some shortcomings that need to be addressed.

Please consider the following comments to improve the content of your manuscript before publication. 

  1. The introduction continues to lack relevant information about the synbiotic being evaluated. They cite a series of beneficial effects attributed to synbiotics but never mention the direct antecedents of the synbiotic used in the study. The definition of a synbiotic (lines 52-54) is still incorrect. They mention in the response letter that they corrected it, but this was not the case.

  1. Section 2.1 "SBT Supplements Preparation" describes the composition of the commercial product in a little more detail; however, it does not indicate how many strains of the Lactobacillus and Bifidobacterium genera the product contains. The concentration of 7.4 x 108 CFU/gram corresponds to the total number of probiotic microorganisms, or is it the amount per probiotic strain? I believe more information should be included in this regard to determine the amount of each microorganism strain being administered.

Please amend the requested comments and submit the revision file.

Author Response

Response to Reviewer 1 Comments

Comment 1: The introduction continues to lack relevant information about the synbiotic being evaluated. They cite a series of beneficial effects attributed to synbiotics but never mention the direct antecedents of the synbiotic used in the study. The definition of a synbiotic (lines 52-54) is still incorrect. They mention in the response letter that they corrected it, but this was not the case.

Response 1: Many thanks to your valuable comment. We have added the definition of a symbiotic and mentioned the direct antecedents of the synbiotic used in the study. Please refer to Lines 52-63 of the revised manuscript.

Comment 2: Section 4.1 "SBT Supplements Preparation" describes the composition of the commercial product in a little more detail; however, it does not indicate how many strains of the Lactobacillus and Bifidobacterium genera the product contains. The concentration of 7.4 x 108 CFU/gram corresponds to the total number of probiotic microorganisms, or is it the amount per probiotic strain? I believe more information should be included in this regard to determine the amount of each microorganism strain being administered.

Response 2: Many thanks to your valuable comment. We have added more information about SBT supplements in Section 4.1. Please refer to Lines 372-380 of the revised manuscript.

Reviewer 2 Report

Comments and Suggestions for Authors

The authors imporved the manuscript as advised. 

Author Response

Response to Reviewer 2 Comments

Comment 1: The authors imporved the manuscript as advised

Response 1: Many thanks your valuable comment.